# Exploring the Impact of Alternative Sources of Dietary Nitrate Supplementation on Exercise Performance

**DOI:** 10.3390/ijms25073650

**Published:** 2024-03-25

**Authors:** Maciej Jędrejko, Karol Jędrejko, Víctor M. Gómez-Renaud, Katarzyna Kała, Bożena Muszyńska

**Affiliations:** 1Department of Pharmaceutical Botany, Faculty of Pharmacy, Jagiellonian University Medical College, Medyczna 9 Str., 30-688 Kraków, Polandkarol.jedrejko@gmail.com (K.J.); k.kala@uj.edu.pl (K.K.); 2Human Performance Laboratory, School of Physical Education, Autonomous University of Nuevo Leon, San Nicolas de los Garza 66455, Mexico; victor.gomezrn@uanl.edu.mx

**Keywords:** *Amaranthus*, spinach, nitrate, alternative dietary nitrate, athletes, exercise performance, recovery

## Abstract

An increase in the level of nitric oxide (NO) plays a key role in regulating the human cardiovascular system (lowering blood pressure, improving blood flow), glycemic control in type 2 diabetes, and may help enhance exercise capacity in healthy individuals (including athletes). This molecule is formed by endogenous enzymatic synthesis and the intake of inorganic nitrate (NO_3_^−^) from dietary sources. Although one of the most well-known natural sources of NO_3_^−^ in the daily diet is beetroot (*Beta vulgaris*), this review also explores other plant sources of NO_3_^−^ with comparable concentrations that could serve as ergogenic aids, supporting exercise performance or recovery in healthy individuals. The results of the analysis demonstrate that red spinach (*Amaranthus* spp.) and green spinach (*Spinacia oleracea*) are alternative natural sources rich in dietary NO_3_^−^. The outcomes of the collected studies showed that consumption of selected alternative sources of inorganic NO_3_^−^ could support physical condition. Red spinach and green spinach have been shown to improve exercise performance or accelerate recovery after physical exertion in healthy subjects (including athletes).

## 1. Introduction

Inorganic nitrate (NO_3_^−^) plays a crucial role in regulating the cardiovascular system in humans, particularly in decreasing blood pressure, improving blood flow, and supporting health management in conditions such as glycemic control in type 2 diabetes. Dietary NO_3_^−^ converted to nitrite (NO_2_^−^), following which NO_2_^−^ is reduced to nitric oxide (NO). The final molecule NO serves as a signaling function in various physiological systems, especially cardiovascular regulation. The nitrogen atom in the NO molecule contains single unpaired electrons, which results in high reactivity and chemical instability. NO demonstrated a short half-life, estimated in the range of 3–5 s. NO in humans can be produced in several ways [1,2].

Human saliva contains high concentrations of NO_3_^−^, which can be reduced to NO_2_^−^ and then converted into NO by oral nitrate-reducing bacteria. It should be noted that NO_2_^−^ is not only converted to NO but can also be further reduced to nitrous oxide (N_2_O) and nitrogen (N_2_) through the denitrification reaction by oral bacteria [3].

NO synthesis is also associated with mitochondrial amidoxime-reducing component (mARC) proteins, which are enzymes that contain molybdopterin. Both isoforms mARC-1 and mARC-2 are responsible for catalyzing an NADH-dependent NO_2_^−^ reduction to NO under hypoxia conditions in the presence of cytochrome b5 (CYB5) and cytochrome b5 reductase (CYB5R). mARC reduces NO_2_^−^ to NO using an electron transfer chain with NADH, CYB5, and CYB5R. In this reaction, molybdenum plays a role as a cofactor, called Mo-cofactor (Moco) [4].

The molybdenum-dependent enzyme family also includes sulfite oxidase (SO), xanthine oxidoreductase (XO), aldehyde oxidase (AO), and nitrate reductase (NR). Furthermore, other enzymes are involved in reducing NO_2_^−^ to NO, such as deoxyhemoglobin, myoglobin, and cytoglobin [5,6].

In the human body, NO serves as a signal transmitter across various physiological systems, particularly in the regulation of the cardiovascular system. Under physiological conditions, an increase in the concentration of vasoconstrict substances (including acetylcholine, bradykinin, angiotensin, endothelin, and adenosine), through an autoregulating mechanism that maintains the body’s homeostasis, induces the production of endothelium-derived relaxing factor (EDRF), which exhibits vasodilator activity. Atherosclerosis, along with the remodeling and degeneration of blood vessel walls, is associated with a reduction in EDRF concentration in the circulatory system. Structurally, EDRF appears in the form of NO and is formed from the amino acid L-arginine in the human body in several ways, involving enzymes such as NO synthase (NOS), which appears in a few isoforms: neuronal (nNOS), endothelial (eNOS), and inducible NOS (iNOS) [1,2].

Nonenzymatic NO production is related to the supply of NO_3_^−^ from the diet, with fresh plants/vegetables or processed plant products such as juices being common sources. Beetroot is particularly renowned for its high NO_3_^−^ content [7].

Athletes, both professional and amateur, strive to achieve the best results in training and competitions. Certain bioactive ingredients contained in food or dietary supplements demonstrate an ergogenic effect, improving human exercise capacity. This support of exercise/endurance performance includes increasing muscular strength and power, accelerating post-exercise recovery through various mechanisms of action, improving oxygen utilization and distribution, improving NO levels, and supporting adenosine triphosphate resynthesis [8,9,10].

According to the International Olympic Committee, NO_3_^−^ supplementation falls under this category “With Good-to-Strong Evidence of Achieving Benefits to Performance When Used in Specific Scenarios” [10].

The Australian Institute of Sport categorizes NO_3_^−^ or beetroot juice supplementation in category A, indicating that these supplements/ingredients “can support or enhance exercise performance” [9].

The Union of European Football Association expert group also issued a statement regarding nutritional intervention in elite football players [11].

Scientific works suggest that the consumption of beetroot in various forms (e.g., juice or powder/extract) correlates with improved physical conditions in selected types of exercise. Certain studies demonstrate that supplying dietary NO_3_^−^ before exertion contributes to improvements in cardiovascular and respiratory parameters, physical endurance, high-intensity intermittent exercise performance, and muscle power or sprinting [12,13].

In addition to beetroot, the high content of NO_3_^−^ was also confirmed in other plants such as red spinach (*Amaranthus* spp.) or green spinach (*Spinacia oleracea*).

The aim of this work was to investigate the impact of alternative dietary NO_3_^−^ supplementation on exercise capacity or recovery in humans, based on available studies conducted on healthy individuals.

## 2. Materials and Methods

The methodology used to gather and assess the clinical studies included a primary initial search on the PubMed database and secondary searches through the Google Scholar database. The timeframe for data collection was from 1 August 2023 to 20 December 2023. The methodology diagram is depicted in Figure 1 [9].

### 2.1. Identification

The strategy used to evaluate the available literature employed the following search terms: ((spinach) OR (amaranthus)) AND ((exercise) OR (performance) OR (athlete) OR (oxygenation) OR (antihypoxic) OR (recovery) OR (endurance) OR (muscles) OR (physical) OR (ergogenic)). The initial search provides 3109 results (*n* = 3109).

### 2.2. Screening

The remaining publications were filtered by the criteria “Article type” with only works available as “Randomized Controlled Trial”, “Clinical trials”, and “Humans” selected, which resulted in 39 records (*n* = 39). Among the remaining publication records, those that were available only as an abstract were excluded, with 38 “Full text” publications (*n* = 38) remaining. Then, the initial search results were further screened manually for relevance. Several publications were excluded based on this analysis (*n* = 31). Finally, seven studies were included (*n* = 7).

### 2.3. Eligibility

The secondary search of PubMed and an additional source, Google Scholar, provided four supplementary records (*n* = 4).

### 2.4. Inclusion

The final review included 11 publications (*n* = 11).

## 3. Results and Discussion

Following the search on PubMed and Google Scholar, a total of 11 records were identified. Among them, eight studies were related to red spinach supplementation, whereas three were associated with green spinach supply. All details about the collected studies are summarized in Table 1.

Supplementation of 1 g of red spinach extract (RSE) over seven days in a group of 12 young recreationally trained participants (aged 21–25 years) increased blood NO_3_^−^ concentrations. Although the intake of RSE improved only the ventilatory threshold (VT), no significant differences between the experimental and placebo groups were observed in other exercise parameters, including peak oxygen uptake (VO_2peak_) and time to exhaustion (TTE) under treadmill exercise test (Bruce protocol) [14].

Moore et al. (2017) demonstrated that a single dose of 1 g of powder RSE (90 mg NO_3_^−^) contributed to increased NO_3_^−^ concentrations in blood in a group of 15 young participants (eight male and seven female participants). NO_3_^−^ levels were measured 60–75 min after intake of RSE. Impact on exercise performance was associated with an increase in VT. However, no significant changes in TTE and VO_2peak_ were confirmed. In this study, RSE derived from the dietary supplement Oxystorm (*Amaranthus dubius* extract standardized to 9% NO_3_^−^) was used [15].

In a study conducted among 11 female football players (aged 19–22 years), acute supplementation with 4.4 g RSE (400 mg NO_3_^−^) from the dietary supplement Spin Boost (Vita Spinach^®^ extract) demonstrated no improvement in power, heart rate (HR), and rate of perceived exertion. However, RSE intake increased plasma NO_2_^−^ and blood lactate (LA) concentrations. In addition, a reduction in fatigue index values was also noted. It has been suggested that higher blood LA concentrations, along with reduced fatigue, may be associated with faster elimination of lactate from muscles [16].

Liubertas et al. (2020) compared acute and short-term supplementation (six days) of 4 g of standardized concentrate from *Amaranthus hypochondriacus* (400 mg NO_3_^−^). A group of 13 male participants (aged 22–24 years) intake RSE in the form of an oat bar. Acute supplementation consisted of taking the bar 1 h before an exercise test, increasing cycling exercise (ICE). During short-term, six-day supplementation, the athletes intake the bar as breakfast. On the last day, the RSE was reapplied 60 min before the ICE test. Short-term supplementation with RSE resulted an increase in peak power in ICE test and boosted VT, likewise maximal oxygen uptake (VO_2max_) values. However, a single acute dose did not result in significant changes in the participants’ exercise capacity [17].

Another study investigated the effect of 15-day supplementation with 4 g of RSE as Oxystorm^®^ (~360 mg of NO_3_^−^) among 11 healthy males (aged 30–42 years). Intake of RSE improved high-intensity exercise tolerance (increasing TTE values). TTE was significantly greater, approximately 19% in the experimental group compared to the placebo. In addition, they reported an increased exhaled NO, likewise reducing blood pressure (BP) in participants [18].

In the case of resistance training (barbell bench press), it was demonstrated that a seven-day supplementation with 2 g of RSE (180 mg NO_3_^−^) from Super Spinach supplement (*Amaranthus dubis* extract) did not improve muscle oxygenation and exercise performance in young resistance-trained males. In this study, it was reported that RSE contributed to a subjective increase in focus and improved “muscle pumps” [19].

Townsend et al. (2022) investigated the long-term supplementation of 2 g of RSE (Super Spinach supplement) among 16 young male Division I baseball players for 11-week off-season trainings. Intake of red spinach contributed to improvements in peak power during the Wingate test, with no significant changes in body composition, strength, and cardiovascular parameters [20].

Mixed results were achieved in a group of 17 young physically active men and women (aged 19–26 years). In this study, one-week supplementation with RSE (Super Spinach supplement) in a dose of 1 g/day before exercise (4 km cycling time trial test) was evaluated. The participants using RSE had a decrease in BP and an improvement in average speed, power, and a reduction in the time to completion (TTC) [21].

Short-term supplementation of fresh green spinach leaves (1 g/kg of body weight) over 14 days was investigated in a group of 20 well-trained male runners. Participants in this study had 21 km to complete. After the run, blood samples were taken for analysis. Runners consuming green spinach had an increased total antioxidant capacity (TAC) post-exercise, a decrease level in the levels of markers of oxidative stress (malondialdehyde, protein carbonyl, and uric acid), as well as a decrease in the levels of markers of muscle damage such as creatine kinase and lactate dehydrogenase [22,23].

Pérez-Piñero et al. (2021) verified the effect of long-term supplementation of green spinach extract 2 g daily (708 mg NO_3_^−^) among a group of 45 adult individuals (37 women and eight men) aged older than 50 years. Participants performed resistance training (three times per week), for 12 weeks. The progression of intensity was chosen individually for each volunteer. The group that consumed green spinach extract demonstrated nonsignificant changes (compared with placebo) in body composition such as a decrease in fat mass and an increase in lean mass. Significant changes mainly occurred in muscle quality and function, e.g., isokinetic and isometric dynamometry. Remarkably, male volunteers who supplemented with green spinach extract demonstrated significant changes in muscle mass and isokinetic or isometric muscle strength compared to female participants [24].

The effectiveness of beetroot intake, particularly in juice form by athletes, in supporting exercise capacity has been confirmed by some studies [25,26,27]. However, selected studies have shown no effect of beetroot consumption on exercise performance [28,29,30]. The reviewed studies in this article show evidence of the positive effects of red spinach standardized extract on improving VT [14,15,17], VO_2_max, and peak power during increasing cycling exercise [17], increasing TTE [18], peak power on the Wingate anaerobic test [20], and decreasing the fatigue index [16]. These findings are consistent with the results of a recent investigation, in which dietary NO_3_^−^ from beetroot supplementation had small but significant positive effects on some performance outcomes during single and repeated bouts of high-intensity exercise [31], as well as endurance, high-power explosive, and high-intensity intermittent exercise [32].

Although NO_3_^−^ has been associated with beneficial effects in the context of physical activity, there are established standards for its acceptable daily intake (ADI). The current ADI for NO_3_^−^ is 3.7 mg/kg b.w. Health risks are associated with the conversion of NO_3_^−^ to NO_2_^−^, which can lead to methemoglobinemia and disrupt proper oxygen distribution. In addition, NO_3_^−^, in reaction with amines, can produce carcinogenic nitrosamines. For comparison, the ADI for NO_2_^−^ was established at 0.07 mg/kg b.w. [33,34,35].

The most recognized natural source of dietary NO_3_^−^ is beetroot (*Beta vulgaris*), containing NO_3_^−^ in the range of 2500 mg/kg. Additionally, apart from NO_3_^−^, beetroot also includes betaine and betanins (betacyanins). Betanins are responsible for the intense red color of beetroot [36].

*Amaranthus spp.*, commonly known as amaranth or red spinach, is a plant that comes from South and Latin America (Peru, Mexico). It is also a popular plant crop in Asia (China, India) and Africa. Amaranth came to Europe by the Spanish during the colonization of South America. There are many species of red spinach, including *A. tricolor*, *A. hypochondriacus*, *A. hybridus*, *A. dubius*, and *A. cruentus*. Depending on the geographical region, the aerial parts of red spinach (leaves and grains) and the underground parts—root/rhizome of amaranth—are used for food purposes. The grains of red spinach are a rich source of proteins, carbohydrates (primary starch), lipids (including linoleic, oleic, and palmitic acids), and numerous bioelements, such as potassium, calcium, iron, and magnesium. Amaranth leaves, on the other hand, are a source of antioxidants such as anthocyanins and carotenoids [37,38].

The NO_3_^−^ content, depending on the amaranth species, ranges from 2800 to 8800 mg/kg (in leaves) [39].

Vegetables, such as *Spinacia oleracea* L. (green spinach), contain NO_3_^−^, betaine, and polyphenols (mainly flavonoids). The high concentration of NO_3_^−^ confirmed in spinach leaves ranges from 900 to 5400 mg/kg [40,41,42]. Depending on the species of spinach, the concentration of polyphenols has been shown to be inversely proportional to the content of NO_3_^−^. Spinach leaves with a high concentration of NO_3_^−^ have a low level of vitamin C and flavonoids [43]. The content of NO_3_^−^ in red spinach and green spinach is presented in Table 2.

Green spinach is also a source of plant steroids such as ecdysterone, which demonstrates anabolic properties. The concentration of ecdysteroids in green spinach is estimated to range from 17.1 to 885 µg/g dry weight. The ecdysterone content is much higher in the leaves than in the roots or seeds [44,45,46].

## 4. Conclusions

Beetroot is not the only valuable source of NO_3_^−^ in the daily diet for physically active persons. There are several other plants that can supply NO_3_^−^ in the daily diet. Some of these, such as red and green spinach, have been shown to support exercise capacity in humans. Further studies with different high-intensity exercise protocols should be considered to evaluate the effects of red spinach and green spinach versus beetroot juice supplementation, to compare their effects on different outcome variables to evaluate whether athletes competing in sports requiring those exercise characteristics can benefit from supplementation with alternative NO_3_^−^ sources.

## Figures and Tables

**Figure 1 ijms-25-03650-f001:**
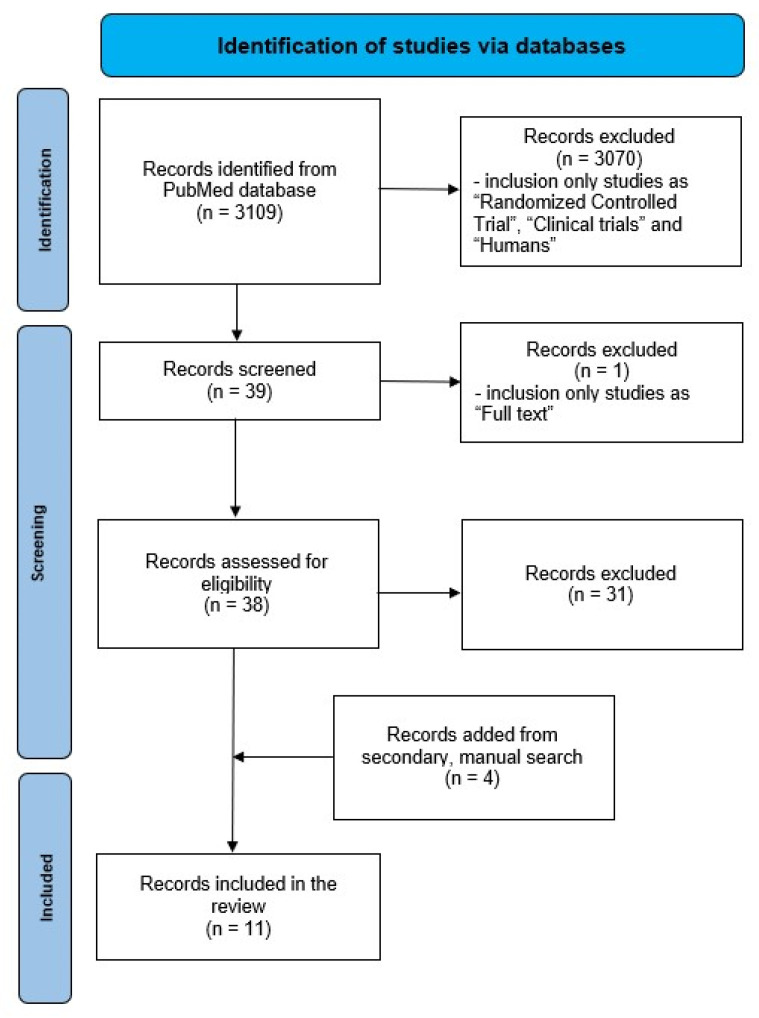
Methodology diagram.

**Table 1 ijms-25-03650-t001:** Effect of dietary nitrate supplementation on exercise performance.

Ingredient	Participants	Dose	Duration Time	Investigated Parameters	Results	References
*Amaranthus* *dubius* standardized extract Placebo	Healthy recreationally trained individuals *n* = 12 Aged 21–25 (6 males and 6 females)	1 g (90 mg NO_3_^−^)	7 days	Blood sample analyzed Treadmill exercise test (Bruce protocol) VT TTE VO_2peak_	↑ NO_3_^−^ concentration ↑ VT ↔ TTE ↔ VO_2peak_	[14]
*Amaranthus* *dubius* standardized extract Placebo	Healthy recreationally trained individuals *n* = 15 Aged 20–28 (8 males and 7 females)	1 g (90 mg NO_3_^−^)	Acute dose	Blood sample analyzed Treadmill exercise test (Bruce protocol) VT TTE VO_2peak_	↑ NO_3_^−^ concentration ↑ VT ↔ TTE ↔ VO_2peak_	[15]
*Amaranthus* *dubius* standardized extract Tomato juice (placebo)	Female soccer players *n* = 11 Aged 19–22	4.4 g (400 mg NO_3_^−^)	Acute dose	Blood sample analyzed Wingate Anaerobic Tests Peak power Cycle ergometer Heart rate (HR) and rate of perceived exertion	↑ NO_2_^−^ and LA concentration ↓ Fatigue index No improvement in peak power, HR, and rate of perceived exertion	[16]
*Amaranthus* *hypochondriacus* standardized extract (9–11%) Oat bar (placebo)	Male volunteers *n* = 13 Aged 22–24	4 g (400 mg NO_3_^−^)	Acute dose	Increasing cycling exercise (ICE) test Peak power Cycle ergometer VT VO_2max_	No significant changes after acute intake Improvement exercise parameters after short-term supplementation	[17]
			6 days		↑ Peak power in ICE test ↑ VT ↑ VO_2max_	
*Amaranthus* *dubius* standardized extract Placebo	Male volunteers *n* = 11 Aged 30–42	4 g (~360 mg NO_3_^−^)	15 days	Cardiovascular measures (BP) Exhaled nitric oxide Cycle ergometer TTE	↓ BP ↑ Fractional exhaled nitric oxide (FeNO) ↑ TTE	[18]
*Amaranthus* *dubius* standardized extract Placebo	Males, resistance-trained *n* = 10 Aged 19–26	2 g (180 mg NO_3_^−^)	7 days	Cardiovascular measures (BP and HR) Muscle oxygen saturation One Repetition Maximum (1RM) test (barbell bench press) Peak power Stroop test Visual analog scales (VAS)	↔ BP and HR No improvement in peak power and muscle oxygenation ↑ Subjective measures—focus and “muscle pump”	[19]
*Amaranthus* *dubius* standardized extract Placebo	Males, college baseball athletes *n* = 16 Aged 19–22	2 g (180 mg NO_3_^−^)	11 weeks	Cardiovascular measures (BP and HR) Body composition One Repetition Maximum (1RM) test (barbell bench press) Wingate Anaerobic Tests	No improvement in body composition, cardiovascular measures, and strength ↑ Peak power in Wingate test	[20]
*Amaranthus* *dubius* standardized extract Placebo	Healthy, recreationally active individuals *n* = 17 Aged 19–26 (8 males and 9 females)	1 g (90 mg NO_3_^−^)	7 days	Cardiovascular measures (BP and HR) Subjective measures (muscle fatigue, perceived exertion) 4 km cycling time trial test Performance measures (TTC, average power, average speed)	↓ BP ↓ TTC ↑ average speed ↑ average power	[21]
Green spinach leaves (*Spinacia oleracea*) Placebo	Males, well-trained runners *n* = 20 Aged 22–24	1 g/kg body weight	14 days	Total antioxidant capacity (TAC) Markers of muscle damage (LDH, bilirubin)	↑ TAC ↓ LDH after physical activity ↓ Bilirubin concentration	[22]
Green spinach leaves (*Spinacia oleracea*) Placebo	Healthy well-trained runners *n* = 20 Aged 22–24	1 g/kg body weight	14 days	TAC Markers of oxidative stress (malondialdehyde, protein carbonyl, uric acid) Markers of muscle damage (creatine kinase)	↑ TAC ↓ Markers of oxidative stress ↓ Markers of muscle damage	[23]
Green spinach (*Spinacia oleracea*) extract Placebo	Adult individuals *n* = 45 Aged 50–75 (8 males and 37 females)	2 g (708.8 mg NO_3_^−^)	12 weeks	Body composition Muscle strength (dynamometry)	↔ Body composition ↑ Isometric and isokinetic strength in males	[24]

Aberrative: RSE—red spinach extract; NO_3_^−^—nitrates; NO_2_^−^—nitrites; VT—ventilatory threshold; VO_2peak_—peak oxygen uptake; VO_2max_—maximum oxygen uptake; ICE—increasing cycling exercise; LA—lactic acid; LDH—lactate dehydrogenase; BP—blood pressure; HR—heart rate; TAC—total antioxidant capacity; TTC—time to completion; TTE—time to exhaustion; ↑—increased; ↓—decreased; ↔—without significant changes.

**Table 2 ijms-25-03650-t002:** Dietary nitrate concentration in vegetables.

Vegetable	Plant Sources	Nitrates Content (mg/kg)	References
*Amaranthus dubius*	Fresh leaves	4100–5200	[39]
*Amaranthus hypohondriacus*	Fresh leaves	4200–7800	[39]
*Amaranthus tricolor*	Fresh leaves	2800–8800	[39]
*Spinacia oleracea*	Fresh leaves	1044.2	[41]
		2170	[40]
		900–5400	[42]

## Data Availability

The data presented in this study are available on request from the corresponding author.

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
