# Peer review of "Exploring the Impact of Alternative Sources of Dietary Nitrate Supplementation on Exercise Performance"

_ijms, 2024, doi:10.3390/ijms25073650_

Round 1

Reviewer 1 Report

Comments and Suggestions for Authors

This review article is entitled "Exploring the impact of alternative sources of dietary nitrate supplementation exercise performance" by Maciej JÄ™drejko et al.  which is probably mistakenly written instead of  "Exploring the impact of alternative sources of dietary nitrate supplementation ON exercise performance"!

It is overall a nicely written manuscript with some minor English language misuse. 

However, the actual aim of this review is not very clear. The Methodology section describes a protocol followed for the inclusion and exclusion of the reported literature, refers to clinical trials and finally stating " The final review included 11 publications". On the other hand, the introduction states that "The objective of this work was to investigate the impact of alternative dietary NO3ˉ  supplementation on exercise capacity or recovery in humans, based on the available studies conducted on healthy individuals." which is a more general description. And finally, after the reporting of the 11 trials, there is a discussion section with references and information on different uses of alternative NO3- sources such as

line 239 a significant increase in hemoglobin levels was observed

line 262 Adding spinach to the diet resulted in a slight increase in lutein levels, whereas the inclusion of tomatoes led to a marked increase in lutein and lycopene concentrations

LIne 278 "A study by Bondonno et al. (2014) demonstrated that consumption of green spinach increased NO3ˉ concentration but had no effect on improving cognitive performance in healthy individuals [50].

The title is therefore not fully descriptive of the content of this review and there is a need of extensive reorganisation of the review purpose and presentation. 

Comments on the Quality of English Language

Some minor English editing is needed.

Author Response

Comments and Suggestions for Authors

This review article is entitled "Exploring the impact of alternative sources of dietary nitrate supplementation exercise performance" by Maciej JÄ™drejko et al.  which is probably mistakenly written instead of  "Exploring the impact of alternative sources of dietary nitrate supplementation ON exercise performance"!

Kindly thank You for valuable tips and recommendations. We reorganised MS in almost all sections. Also, updated mistake in the title.

It is overall a nicely written manuscript with some minor English language misuse. However, the actual aim of this review is not very clear. The Methodology section describes a protocol followed for the inclusion and exclusion of the reported literature, refers to clinical trials and finally stating " The final review included 11 publications". On the other hand, the introduction states that "The objective of this work was to investigate the impact of alternative dietary NO3ˉ  supplementation on exercise capacity or recovery in humans, based on the available studies conducted on healthy individuals." which is a more general description. And finally, after the reporting of the 11 trials, there is a discussion section with references and information on different uses of alternative NO3- sources such as

line 239 a significant increase in hemoglobin levels was observed

line 262 Adding spinach to the diet resulted in a slight increase in lutein levels, whereas the inclusion of tomatoes led to a marked increase in lutein and lycopene concentrations

LIne 278 "A study by Bondonno et al. (2014) demonstrated that consumption of green spinach increased NO3ˉ concentration but had no effect on improving cognitive performance in healthy individuals [50].

The title is therefore not fully descriptive of the content of this review and there is a need of extensive reorganisation of the review purpose and presentation.

The Abstract and Introduction have been updated.

Selected sentences are updated in Abstract:

-"help enhance exercise capacity in healthy individuals (including athletes)".

-“Red spinach and green spinach have been shown to improve exercise performance, or accelerate recovery after physical load in healthy subjects (including athletes)”.

Now it should overlapping with the aim of this review in Introduction section: "The aim of this work was to investigate the impact of alternative dietary NO3ˉ supplementation on exercise capacity or recovery in humans, based on the available studies conducted on healthy individuals".

Discussion section was updated, and removed other plants. Saved only red spinach and green spinach. Provided new content about comparison of studies with red spinach vs beetroot juice:

“The effectiveness of beetroot intake, particularly in juice form by athletes, in supporting exercise capacity has been confirmed by some studies [25–27]. However, selected studies have shown no effect of beetroot consumption on exercise performance [28–30]. The reviewed studies in this article showed evidence of the positive effects of red spinach standardized extract on the improvement of VT [14, 15, 17], VO2max and peak power of the increasing cycling exercise [17], increase of TTE [18] and peak power on Wingate anaerobic test [20], and decreasing the fatigue index [16]. These findings are consistent with the results of a recent investigation in which dietary NO3ˉ from beetroot supplementation had small but nevertheless significant positive effects on some performance outcomes during single and repeated bouts of high-intensity exercise [31], as well as endurance, high-power explosive, and high-intensity intermittent exercise [32].”

Due to MS needed a major rebuild, we didn't have enough time to prepare a well discussion. We gently ask for further analysis, especially a comparison of the studies on red spinach vs beetroot. If the reviewers decide that a second round of review is required, we are ready to make further improvements to maintain the high standard of the IJMS journal.

Reviewer 2 Report

Comments and Suggestions for Authors

Dear Authors,

In this article, the authors study the role of nitrate, through its transformation into NO, in exercise enhancement, analyzing studies in which vegetables with high nitrate content have been given to various types of athletes. I believe the idea is good; however, there are significant deficiencies in both formal and scientific writing that are important and prevent, in my opinion, the publication of the article in its current state until they are addressed. I advise the authors on how to do so.

Mayors

* The organization of the article is incorrect. On one hand, it states that it will present the results and discussion together in a single section, and then it starts a separate discussion.

* L31: The authors do not explain that it is actually nitrite that is reduced to NO, not nitrate. The authors should comment on how this occurs, and the role of oral bacteria in this process. 10.1159/000529162

*L34-L37: This sentence doesn't make sense; the systems mentioned by the authors synthesize NO but from arginine, not from nitrate. The systems that synthesize NO from nitrate are different, as indicate next.

*L48: “Nonenzymatic NO production” The authors have forgotten to include that in humans, an enzymatic system called mARC (molybdoenzymes) capable of reducing nitrite to NO was discovered in 2014, as reported in https://doi.org/10.1074/jbc.M114.555177.  

*And in addition to mARC, the mitochondrial electron transport chain, sulfite oxidase,….. and other mechanisms have also been demonstrated to be capable of reducing nitrite to NO, as documented in this review DOI 10.3390/molecules23123287. Due to its significance, this mechanisms and work should also be and cited in this review.

*Also, the reaction of nitrite with oxyhemoglobin is well established and generates nitrate and methemoglobin. Could you also address this aspect? And if something is known about the health implications?

 *L99: “3 supplementary records (n = 4).” 3 or 4?

*L189: It is important that the authors indicate the governmental regulatory limits of dietary intake nitrate, but also nitrite.

L143: “we demonstrate that a 7-day” We?

*I don't understand what 'Not covered on PubMed' means in Table 1

* I observe that the discussion is very deficient, primarily because the authors in the discussion should focus on the 11 studies shown in the results and relate them to each other, pointing out similarities and differences. This aspect needs to be clearly improved. And secondary because  it's too long, and it covers too many topics unrelated to the exercise. The authors should make an effort to review such references and remove them; I fail to see how they contribute to the topic at hand. Clear example is this paragraph here: L241-L247 and L278-L280, but there are more, as I said the authors must identify and remove them.

* I don't understand why in the discussion the authors spend so much time commenting on various aspects of Rheum rhabarbarum, Asparagus spp., Musa acuminata, Brassica oleacera,... and others, if their search results didn't yield any information on these and the exercise. I don't see how it contributes to the paper's topic; it should be minimized or removed altogether.

*In Table 2, as previously mentioned, remove the species that are not relevant to the paper's topic, the exercise.

Comments on the Quality of English Language

Fine

Author Response

Kindly thank You for valuable tips and recommendations. We reorganised MS in almost all sections.

All reponses in the file.

Due to MS needed a major rebuild, we didn't have enough time to prepare a well discussion. We gently ask for further analysis, especially a comparison of the studies on red spinach vs beetroot. If the reviewers decide that a second round of review is required, we are ready to make further improvements to maintain the high standard of the IJMS journal.

Round 2

Reviewer 1 Report

Comments and Suggestions for Authors

No additional work is needed

Comments on the Quality of English Language

minor editing is required

Reviewer 2 Report

Comments and Suggestions for Authors

Dear Authors,

I believe the authors have adequately addressed all of my comments and suggestions, and I accept the paper in its current version